# Development of a Graphene Oxide-Based Aptamer Nanoarray for Improved Neutralization and Protection Effects Against Ricin

**DOI:** 10.3390/pharmaceutics16111455

**Published:** 2024-11-14

**Authors:** Huafei Li, Yanwen Ai, Yanjin Wu, Ruyu Fan, Yuan Tian, Shuangqun Chen, Wei Wan, Cong Wu

**Affiliations:** 1School of Lifesciences, Shanghai University, 333 Nanchen Road, Shanghai 200444, China; 2Clinical Research Unit, The First Affiliated Hospital of Navy Medical University (Changhai Hospital), 168 Changhai Road, Shanghai 200433, China; 3Department of Orthopedic Oncology, Spinal Tumor Center, Changzheng Hospital, Navy Medical University, 415 Fengyang Road, Shanghai 200003, China

**Keywords:** Ricin, aptamer, nanoarray, toxin neutralization, biological defense

## Abstract

**Background/Objectives**: Ricin’s high toxicity and potential as a bioweapon underscore the need for effective antidotes. Monoclonal antibodies, though effective, are limited by complex production. This study aimed to develop a graphene oxide-based aptamer nanoarray (ARMAN) for improved neutralization and protection against ricin. **Methods:** High-affinity aptamers targeting ricin’s RTA and RTB subunits were selected using SELEX technology and conjugated to graphene oxide (GO) via click chemistry. ARMAN’s characteristics, including morphology, stability, and biosecurity, were assessed. Its performance was evaluated in terms of affinity for ricin, neutralization capacity, and therapeutic effects in cellular assays and a mouse model of ricin poisoning. **Results**: ARMAN exhibited a uniform morphology with an average particle size of 217 nm and demonstrated significantly enhanced affinity for ricin compared to free aptamers. ARMAN showed rapid and effective neutralization ability, significantly increasing cell viability in BEAS-2B, GES-1, and HL7702 cell lines exposed to ricin. In vivo, ARMAN treatment led to a notable prolongation of survival in ricin-poisoned mice, highlighting its potential for both pre- and post-exposure treatment. These findings indicate that ARMAN not only neutralizes ricin effectively but also provides a therapeutic window for treatment. **Conclusions**: ARMAN’s superior binding affinity, serum stability, biocompatibility, and broad therapeutic efficacy make it a promising new antidote against ricin poisoning. This study’s findings represent significant progress in the development of rapid-response antidotes, with ARMAN offering a potential solution for both military and civilian emergency response scenarios.

## 1. Introduction

Ricin toxin (RT) is a potent protein toxin derived from the castor bean (*Ricinus communis*), which belongs to the type II ribosome inactivating protein (RIP) family and has captured researchers’ attention due to its exceptional toxicity and potential as a biochemical weapon [1]. RT holotoxin consists of two distinct chains, which are linked by disulfide bonds; the Ricin toxin A-chain (RTA) possesses *N*-glycosidase activity, which can depurinate a specific adenine base in the 28S rRNA of the large ribosomal subunit, effectively halting protein synthesis and leading to cell death, while the Ricin toxin B-chain (RTB) is responsible for cell binding and internalization [2]. By binding either the surface of the galactosyl moieties or mannose receptors, the RTB facilitates its entry into eukaryotic cells, with the capacity to attach 106 to 108 Ricin molecules per cell surface [3].

RT can cause death to poisoned individuals through oral ingestion, intramuscular injection, or even short-term inhalation. The severity of its effects differs according to the exposure route, with inhalation proven to be more potent compared to oral ingestion [4,5]. Specifically, the median lethal dose (LD50) for the inhalation of RT is between 3 and 5 mg/kg, whereas the oral LD50 is 20 mg/kg. This discrepancy may be attributed to RT’s relatively large molecular size and its degradation within the gastrointestinal (GI) tract [5]. Due to its exceptional toxicity, stability, and accessibility, the United States Department of War identified RT as a potential weapon in 1918, assigning it the name “W compound”. Both Britain and the United States subsequently developed a “W bomb”, conducting tests throughout World War II [5]. Despite never being used in actual combat, RT continues to pose a significant threat to public safety and is currently being weaponized by several countries and terrorist groups [6,7]. Presently, RT has been categorized as a chemical/biological weapon under the Chemical Weapons Convention (CWC) and the Biological and Toxin Weapons Convention (BTWC) [6].

RT remains a substance for which there is currently no optimal antidote. Among various strategies aimed at detoxifying, the use of neutralizing monoclonal antibodies (mAbs) stands out as a potential approach [1].

Over the past few decades, significant advancements have been made in the development and testing of mAbs specifically designed to neutralize RT and enhance survival rates following exposure. Since the initial studies in the 1980s, in vitro experiments have successfully achieved inhibitory concentrations for the RTA subunit, resulting in encouraging outcomes with high survival rates observed in various animal species, including mice and monkeys, exposed to lethal RT through different routes of toxin uptake [7,8,9,10,11]. While the majority of research has concentrated on targeting RTA, progress has also been made in the development of mAbs against RTB [1,11,12]. Furthermore, researchers have investigated the use of combination therapies, or cocktail therapies, targeting both RTA and RTB. These combined treatments have demonstrated impressive detoxifying effects in both in and ex vivo experiments, often yielding superior outcomes compared to separate anti-RTA and anti-RTB mAb treatments [11,13,14,15].

Despite the promising neutralization effects of mAbs in laboratory settings, no mAbs have yet been approved for Ricin intoxication until now [13]. Besides, their complex preparation process and extended production time make them impractical for swift deployment in military emergencies or critical response situations. Consequently, the development of novel detoxifying agents holds immense strategic and tactical significance, particularly in such urgent circumstances.

Aptamers are single-stranded DNA or RNA molecules (ranging from 50 to 100 nucleotides) that have been refined through the advanced SELEX (Systematic Evolution of Ligands by Exponential Enrichment) technology [16,17]. With the ability to fold into diverse secondary structures and further assume complex three-dimensional conformations, aptamers can effectively interact with their intended targets with high selectivity [18]. When compared to mAbs, aptamers offer several advantages, including excellent reproducibility, ease of modification, small size, reduced toxicity, high stability, and low molecular weight [19,20]. Additionally, aptamers can target a wide range of molecules, including proteins, viruses, cells, and even metal ions [20,21]. Due to these unique characteristics, aptamers have been developed as therapeutic agents in clinical settings. Notably, Pegaptanib was the first aptamer-based drug approved by the Food and Drug Administration (FDA) in 2004 for the treatment of age-related macular degeneration (AMD) [22].

Research has shown that aptamers can be effective tools for detecting and neutralizing the toxic effects of Ricin. Several studies have reported established aptamer sequences for either Ricin A or B chains. For instance, Qi et al. identified an RNA aptamer specific to the Ricin A-chain, demonstrating its potential for use in sensors and diagnostics [23]. Additionally, Lamont et al. developed a DNA aptamer targeting the Ricin B chain, showing promise in the detection of Ricin in various biological samples [24]. Despite these advances, most existing aptamer research has primarily centered on the diagnosis of Ricin exposure rather than the development of therapeutic agents aimed at neutralizing the toxin. This gap highlights the need for novel aptamer sequences that can provide effective solutions for Ricin detoxification.

In this study, we successfully generated multiple high-affinity aptamers that specifically target the RTA and RTB subunits through the SELEX process. Furthermore, we developed a novel aptamer nanoarray by conjugating two aptamers, each with a relatively high affinity for RTA or RTB, onto a two-dimensional graphene oxide (GO) surface. This innovative methodology resulted in the creation of a multifunctional aptamer nanoarray, termed the Anti-ricin Multifunctional Aptamer Nanoarray (ARMAN). Our experimental findings demonstrate that the ARMAN exhibits exceptional neutralization and protective effects against Ricin toxin (RT) in both in vivo and in vitro investigations.

## 2. Materials and Methods

### 2.1. Cells and Ricin Toxin

Normal human alveolar epithelial cell line (BEAS-2B), human gastric mucosal epithelial cell line (GES-1), and human liver cell line (HL7702) were obtained from the American Type Culture Collection (ATCC, Manassas, VA, USA). Cells were incubated in Dulbecco’s modified eagle medium (DMEM) supplemental with 10% (*v*/*v*) fetal bovine serum (FBS, Gibco, Waltham, MA, USA) and 1% penicillin/streptomycin (Thermo Fisher Scientific, Waltham, MA, USA) in a humidified incubator at 37 °C with 5% CO_2_, respectively. Ricin was prepared by the Academy of Military Medical Sciences of the People’s Liberation Army (PLA) (Beijing, China).

### 2.2. Animals

Four-week-old female ICR mice were obtained from Slack Laboratory Animals Co., Ltd. (Shanghai, China) and housed in specific-pathogen-free (SPF) conditions. All the experiments on live mice were approved by the Committee on Animals of Shanghai University (Shanghai, China) and the First Affiliated Hospital of Navi Medical University (Shanghai, China).

### 2.3. Generation of RTA/B-Coupled Magnetic Microbeads

Amine functionalized acid aptamers were conjugated to Carboxyl Derivatized magnetic beads (Thermo Fisher Scientific, Carlsbad, CA, USA) using the EDC/NHS crosslinking method. First, beads were suspended in 0.1 M 2-(*N*-morpholino) ethanesulfonic acid (MES) buffer (pH 5.0) and activated with EDC (10 mg/mL, Aladdin Biochemical Technology Co., Ltd., Shanghai, China) and NHS (10 mg/mL, Aladdin Biochemical Technology Co., Ltd., Shanghai, China) for 15 min at room temperature. Following activation, the beads were washed three times with MES buffer to remove excess EDC/NHS. The amine-functionalized aptamer (1 µM) was then added to the activated beads and incubated for 2 h at room temperature to allow covalent bonding. After coupling, the beads were washed with PBS to remove any unbound aptamer.

### 2.4. Aptamer Selection by SELEX (Figure 1)

High-affinity aptamers were screened utilizing the SELEX method as described in previous publications [25,26]. Two DNA aptamer libraries were synthesized by Sangon Biotech Co. Ltd. (Shanghai, China). Each library sequence comprises a fixed sequence at both the 5′ and 3′ ends, with a 20-base random sequence in the middle (Table 1). Prior to each selection round, the DNA sample is denatured at 95 °C and swiftly cooled on ice for 5 min as a preconditioning step. For the selection of RTA-specific aptamers, positive selection is carried out using RTA-coupled magnetic microbeads (Thermo Fisher Scientific, Carlsbad, CA, USA), followed by negative selection with empty microbeads for a total of 15 rounds. It should be noted that negative selection was not performed for the initial five rounds. During each round, aptamers are eluted from the target beads by heating the suspension to 95 °C while gently vortexing for 10 min. To prepare the aptamers for the subsequent selection round, the DNA sample is purified via agarose gel extraction. The pooled DNA then undergoes PCR amplification and is digested with Lambda exonuclease (Thermo Fisher Scientific, Carlsbad, CA, USA) to generate a single-stranded aptamer. The selection process for RTB specific aptamers is conducted in a similar manner. The selected aptamers are then sequenced, and their binding affinity and biological function are assessed and compared.

**Figure 1 pharmaceutics-16-01455-f001:**
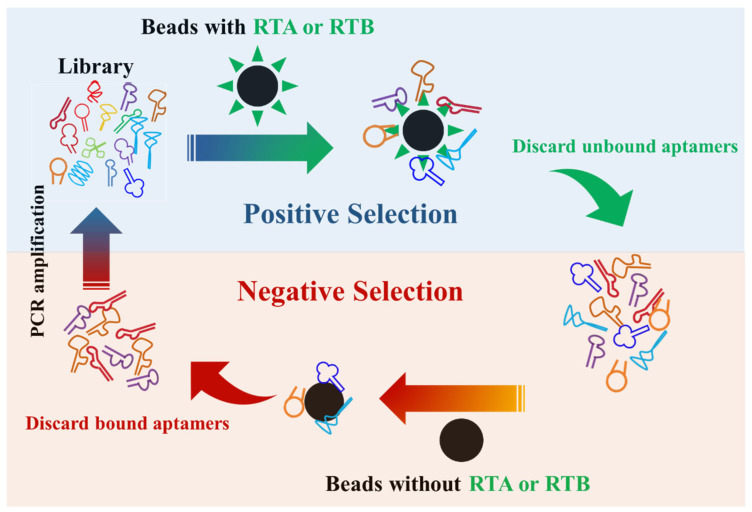
Schematic diagram of the SELEX screening for aptamers targeting RTA and RTB. The red and blue symbols represent RTA and RTB specific aptamers, respectively.

### 2.5. Determination of Aptamer Affinity

The obtained aptamers were labeled with Alexa Fluor 555 and then co-incubated with magnetic beads of RTA or RTB, respectively. The mean fluorescence intensity (MFI) of aptamer conjugated beads was then analyzed by a Flow Cytometer (FCM, CytoFLEX S, Beckman Coulter, San Jose, CA, USA).

### 2.6. Preparation and Characterization of the ARMAN

Click chemistry was employed to pair nanosized graphene oxide (GO) (Ashine new carbon material Changzhou Co., Ltd., Changzhou, Jiangsu, China) with aptamers [26,27]. Briefly, a volume of 10 mL of GO dispersion (0.5 mg/mL in Milli Q water) was gradually introduced to a mixture consisting of 10 mL of HNO_3_ (70%) and 10 mL of H_2_SO_4_ (95%). The resulting mixture was stirred in a water bath maintained at 60 °C for 1 h and subsequently cooled to room temperature. Following this, 500 mL of deionized water was slowly added to achieve dilution. The resultant solution was filtered three times, and the carboxylated GO was collected after freeze-drying. The carboxylated GO was then redispersed in 10 mL of phosphate-buffered saline (PBS) buffer at a concentration of approximately 2 mg/mL and subjected to ultrasonication until the supernatant appeared light brown with no visible solid particles (approximately 30 min). This supernatant was collected as the carboxylated graphene oxide solution (GO-COOH).

Subsequently, GO-COOH was diluted with PBS to achieve a concentration of 1 mg/mL. EDC and NHS were added to achieve a final concentration of 10 mg/mL, which was activated by shaking at room temperature for 15 min. Equal amounts (40 mM) of RTA and RTB targeting aptamers, both modified with amino groups at their 5′ ends, were then added to the solution. After shaking at room temperature for 2 h, free EDC, NHS, and unbound aptamers were removed by ultrafiltration at 4 °C. The resulting ARMAN solution, with a final aptamer concentration ranging from approximately 20 to 29 mM across different batches, was prepared and stored at 4 °C for future use.

The Zeta potential, hydrodynamic diameter, and size distribution of the ARMAN were analyzed using a ZetaSizer (Nano-ZS, Malvern Instruments, Malvern, UK). To observe the microscopic morphology of the ARMAN, transmission electron microscopy (TEM) and atomic force microscopy (AFM, Bruker Dimension ICON, Berlin, Germany) were employed. For TEM specimens, a droplet containing the ARMAN was placed on Holey Carbon Coated Copper 200 mesh Grids (Ted Pella, Redding, CA, USA). After airdrying, the samples were examined on a transmission electron microscope (Emesis Veleta G3, EmCrafts Co., Ltd., Seoul, Republic of Korea) at 80 kV. For the AFM analysis, 10 μL of the ARMAN stock solution was placed onto a freshly cleaved mica strip for 10 min, then gently rinsed and airdried. The samples were mounted onto the XY scanning station and observed using an AFM (Autoprobe CP Research, Veeco Instruments Inc., New York, NY, USA). The structure and spectral characteristics of the ARMAN and its counterparts were measured using a Fourier transform infrared spectrometer (FTIR, Nicolet AVATAR 370, Thermo Scientific, Waltham, MA, USA). To ensure accurate FTIR measurements, GO and ARMAN samples were first thoroughly dried. This was done by placing the samples in a vacuum oven at 40 °C for 12 h to remove any residual moisture, which could interfere with the FTIR spectra by introducing water-related peaks.

### 2.7. Equilibrium Dissociation Constant (Kd) Value Determination

Magnetic beads, which had been coupled with RTA and RTB, were washed using pre-cooled PBS. Subsequently, they were incubated separately with fluorescence-labeled aptamer or ARMAN at varying concentrations at 37 °C for 1 h. Following incubation, the beads were washed again with PBS, and the MFI was measured using an FCM (CytoFLEX S, Beckman Coulter, San Jose, CA, USA). Based on the concentration MFI curve obtained, the equilibrium dissociation constant (Kd) was analyzed using IBM SPSS Software (Version 29.0.2).

### 2.8. Off-Rate Measurement

Magnetic microbeads, which had been coupled with RTA and RTB, were washed with precooled PBS and then incubated with fluorescence-labeled aptamer or ARMAN at a concentration of 400 nM in the dark at 37 °C for 1 h. Following incubation, the beads were washed and resuspended in PBS containing 10% heat-inactivated FBS. The resuspended beads were then incubated at 37 °C with 5% CO_2_. At various time points, aliquots of the microbeads were taken and washed, and the MFI was assessed using an FCM (CytoFLEX S, Beckman Coulter, San Jose, CA, USA). The proportion of aptamers remaining on the surface of the magnetic beads was subsequently calculated according to the following formula:$$\%\;\mathrm{Initial}\;\mathrm{Binding}=\frac{{\mathrm{MFI}}_{\mathrm{samples}}-{\mathrm{MFI}}_{\mathrm{NT}}}{{\mathrm{MFI}}_{0h}-{\mathrm{MFI}}_{\mathrm{NT}}}\times 100\%$$

### 2.9. Serum Stability Evaluation

To assess the serum stability, a solution of 50% (*v*/*v*) FBS in DMEM was utilized as an in vitro serum model, aiming to simulate the in vivo environment. Samples of the ARMAN and GO were then individually mixed with this serum model and incubated at 37 °C for 5 days. The size distribution profile of each sample was evaluated daily using DLS, and comparisons were made to assess any changes over time.

### 2.10. Biocompatibility Evaluation

For the in vitro evaluation, varying concentrations of the ARMAN and free aptamers were added to suspended red blood cells (RBCs) that had been isolated from mouse orbital blood and maintained at room temperature. Milli-Q water and PBS buffer were used as the positive and negative controls, respectively. Following a 3 h incubation period, the cells were centrifuged at 10,000× *g* for 5 min, and the supernatant was collected and transferred to a 96-well plate. The absorbance (Ab) of the supernatant at 540 nm was then measured, and the hemolysis rate was calculated using the following formula:$$\mathrm{Hemolysis}\;\mathrm{rate}\;\left(\%\right)=\frac{{\mathrm{Ab}}_{\mathrm{sample}}-{\mathrm{Ab}}_{\mathrm{negative}\;}}{{\mathrm{Ab}}_{\mathrm{positive}}-{\mathrm{Ab}}_{\mathrm{negative}}}\times 100\%$$

For the in vivo evaluation, healthy 8-week-old female ICR mice were randomly divided into five groups (3 mice per group) and administered with tail vein injections of the ARMAN and its counterparts, each containing an equivalent aptamer amount of 40 μg/kg. After 24 h, blood samples were collected from the mice using the orbital blood sampling method for blood routine and biochemical examinations. Results were then compared among groups.

### 2.11. In Vitro Protective Effects Evaluation

BEAS-2B, GES-1, and HL7702 cells were seeded into 96-well plates at a density of 5 × 10^3^ cells per well and treated with equal amounts of RT. The cells were also incubated with RTA Aptamer (400 nM), RTB Aptamer (400 nM), a combination of RTA + RTB Aptamers (1:1, total amount of 400 nM), and ARMAN (total aptamer amount of 400 nM) for 36 h. After the incubation period, 10 µL of Cell Counting Kit-8 (CCK-8, Abcam, Berlin, Germany) was added to each well, and the plates were incubated for an additional hour in the dark. A microplate reader was then used to measure the absorbance (Ab) at 450 nm, and cell viability was calculated using the following formula:$$\mathrm{Cell}\;\mathrm{viability}\;\left(\%\right)=\frac{{\mathrm{Ab}}_{\mathrm{Sample}}-{\mathrm{Ab}}_{\mathrm{Blank}}}{{\mathrm{Ab}}_{\mathrm{Control}}-{\mathrm{Ab}}_{\mathrm{Blank}}}$$
where Ab_Blank_ are the absorbance values of the culture medium without cells.

### 2.12. In Vivo Protective Effects Evaluation

Healthy female ICR mice aged 8 weeks were randomly assigned to five groups, with ten mice in each group. Each mouse was injected with Ricin at a dose of 40 μg/kg. For the detoxification experiment, mice received tail vein injections of the ARMAN and its counterparts, each containing an equivalent aptamer amount of 40 μg/kg, at 15 min, 1 h, and 2 h after Ricin administration. In contrast, for the prevention–detoxification experiment, the ARMAN and its counterparts were injected 15 min before Ricin administration, as well as 1 and 2 h after. The mice were closely monitored daily for vital signs until their natural death within 14 days. All mice were killed by cervical vertebrae dislocation on day 14.

### 2.13. Statistical Analysis

The experimental results were statistically analyzed using IBM SPSS Software (Version 29.0.2). For the comparison of means between the two groups, a non-paired *t*-test was utilized. When comparing means among three or more groups, a one-way ANOVA was employed. Difference was considered significant at a *p* value of less than 0.05.

## 3. Results

### 3.1. Successful Isolation of Aptamers Specifically Binding to RTA and RTB

In this study, we conducted 15 rounds of SELEX to identify aptamers that specifically bind to RTA or RTB (Figure 1). To track the enrichment of specific aptamers in the secondary libraries after each screening round, we incubated RTA or RTB coupled beads with Alexa Fluor 555 labeled parent DNA libraries and secondary libraries from the 5th, 10th, and 15th rounds. As depicted in Figure 2A, FCM results revealed that as the number of screening rounds increased, the number of aptamers binding to RTA or RTB in the aptamer library gradually rose, indicated by an enhanced MFI. Additionally, the distribution of fluorescence intensity gradually shifted towards a normal distribution. By the conclusion of the 15th round (red curve), the final aptamer library exhibited the strongest binding affinity to the corresponding beads, and the fluorescence intensity distribution curve was relatively uniform. This suggests that after 15 rounds of selection, we successfully obtained an aptamer pool enriched with ssDNA sequences that selectively recognize RTA or RTB. Through sequencing and binding affinity validation, we identified two aptamers targeting RTA and RTB with relatively high avidity, respectively (Figure 2B). The aptamer sequences are as follows: RTA Aptamer—5′-AGAGC GTAGG TTCGC TCGGG AACGG AGTGG TCCGT TATTA ACCAC TATTT GAACC TACC, RTB Aptamer—5′-ACACC CACCG CAGGC AGACG CAACG CCTCG GAGAC TAGCC, and their secondary structures were predicted by simulation analysis using the mfold Web Server online analytical tool (https://www.rna.albany.edu/ (accessed on 30 May 2024)), with the results showing in Figure 2C.

### 3.2. Successful Fabrication of the ARMAN

Previous studies have shown that cocktail therapies targeting both RTA and RTB exhibit superior detoxifying effects compared to separate treatments for either RTA or RTB [11,13,14,15].

To achieve this combined detoxification effect, an innovative Anti-ricin multifunctional aptamer nanoarray (ARMAN) was developed, which was accomplished by cross-linking the two aforementioned aptamers to the surface of GO using click chemistry technology, with the schematic diagram shown in Figure 3A. DLS results indicated that the hydrodynamic radius of the ARMAN and unconjugated GO was 216.8 ± 14.8 nm and 174.4 ± 11.2 nm, respectively (Figure 3B). TEM and AFM images (Figure 3C,D) confirmed that the ARMAN had a high dispersion with an average particle size of approximately 200–240 nm, consistent with the DLS findings. Besides, the AFM results also illustrate that the thickness of the ARMAN was around 6 nm. FTIR spectroscopy (Figure 3E) revealed that both the ARMAN and carboxylated GO (GO-COOH) were rich in hydroxyl groups, indirectly indicating their good water solubility. However, indistinct differences between the GO-COOH and ARMAN were found in the FTIR spectra. This can be attributed to the fact that the FTIR characterization of amide bond formation can be challenging due to peak overlaps with existing functional groups. Specifically, both carboxyl and amide groups exhibit absorption bands in similar regions, complicating spectral differentiation. Despite these limitations in the FTIR spectra, the binding between the aptamers and GO-COOH may still be inferred indirectly by size or morphological alterations. Besides, Figure 3F shows that unmodified GO had a negative Zeta potential of approximately −4.40 ± 0.97 mV, which decreased to −10.68 ± 1.56 mV after carboxylation modification (GO-COOH). Additionally, after being linked with aptamers, the Zeta potential of the ARMAN shifted to −8.39 ± 0.92 mV. The observed Zeta potential shift following aptamer binding can be explained by a dual effect. First, the formation of amide bonds between the carboxyl groups and the amino groups on the aptamer results in partial neutralization of the negative charge associated with the COOH groups. This reduces the overall surface charge density. Secondly, the aptamer itself has a negatively charged phosphate backbone (RTA: −1.70 ± 0.46 mV, RTB: −1.38 ± 1.17 mV), which contributes residual negative charge to the ARMAN structure. This combined effect leads to the observed Zeta potential shift, indirectly confirming the successful linkage between aptamers and GO.

### 3.3. Favorable Serum Stability and Excellent Binding Affinity of the ARMAN

Favorable serum stability is a crucial characteristic for an ideal nanosized drug formulation, especially when intended for intravenous administration (I.V.) [28,29]. To assess the stability of the ARMAN, we utilized DMEM containing 50% FBS (*v*/*v*) as an in vitro serum model, consistent with our previous publications [30]. As depicted in Figure 4A, the ARMAN exhibited superior stability in the serum model compared to unbound GO, with no significant change in average particle size observed over the 5 day experimental period.

The FCM results, presented in Figure 4B, revealed that both the RTA and RTB aptamers demonstrated considerable binding affinity to RT binding microbeads, with dissociation constants (Kd) of 106.2 ± 12.15 nM and 78.57 ± 9.0 nM, respectively. Notably, the ARMAN exhibited even better affinity, with a Kd of approximately 71.48 ± 5.4 nM. The exceptional binding avidity of the ARMAN was further evaluated through a binding off-rate assessment. As shown in Figure 4C, approximately 40.12 ± 6.67% of the ARMAN remained bound to RT conjugating microbeads after 48 h, compared to 14.10 ± 6.2% for RTA aptamer (*p* = 0.000), 19.10 ± 4.78% for RTB aptamer (*p* = 0.000), and 15.40 ± 3.20% for the combination of RTA and RTB aptamers (*p* = 0.000). These findings indicate that the ARMAN has a significantly reduced “off-rate” compared to the free parental aptamers, which may contribute to durable neutralization and antidotal effects against RT.

### 3.4. Excellent Biocompatibility of the ARMAN

Hemocompatibility evaluation was accomplished by incubating mouse erythrocytes with the ARMAN and its counterparts at gradient concentrations. As depicted in Figure 5A, both the free and GO crosslinked aptamers exhibited moderate hemolysis rates (<5%) at concentrations up to 800 nM.

During clinical application, the primary side effects of antidotes are often attributed to their dose-dependent hematotoxicity, hepatotoxicity, and nephrotoxicity. To evaluate these potential toxicities, we measured blood routine and biochemical indexes in plasma samples collected from healthy mice 24 h after tail vein injection of the ARMAN and its counterparts. As shown in Figure 5B–H, there were no significant alterations in the counts of white blood cells (WBC), RBC, or platelets (PLT), nor the levels of creatinine (CREA) and urea nitrogen (BUN). Although there was a slight increase in blood alanine transaminase (ALT) and aspartate transaminase (AST) levels in mice treated with the ARMAN, all values remained within the normal range, indicating minimal systemic toxicity of the ARMAN and the aptamers in vivo.

### 3.5. Exceptional Protective Effects of the ARMAN Against RT In Vitro

To assess the protective efficacy of the ARMAN and free aptamers against RT, CCK-8 assays were conducted to measure the viability of cells exposed to RT and various antidotes. As depicted in Figure 6A–C, 400 nM of free aptamers (RTA + RTB) exhibited a moderate inhibitory effect on the cytotoxicity induced by RT across all tested concentrations ranging from 0.125 to 8 ng/mL (*p* < 0.05). Notably, an equivalent amount of the ARMAN demonstrated superior protective effects compared to its free aptamer counterparts (*p* < 0.05). This trend was consistently observed in all three cell lines tested, including normal human alveolar epithelial cells (BEAS-2B), human gastric mucosal epithelial cells (GES-1), and human liver cells (HL7702).

Subsequently, the aforementioned cells were incubated with 4 ng/mL of Ricin toxin, and 400 nM of free aptamers and ARMAN were used for detoxification. The cell viability ratios of each treatment group to the control group were calculated and compared. As shown in Figure 6D–F, the group treated solely with the ARMAN exhibited no significant difference in cell viability compared to the control group. RT exhibited strong cytotoxicity against all three cell lines, with BEAS-2B and GES-1 cells showing a decreased viability of approximately 40%, and HL7702 cells of 25%. While free RTA or RTB aptamers displayed modest detoxifying effects, their combination significantly enhanced cell viability to 55–70%. In contrast, the ARMAN demonstrated the most remarkable detoxifying effect among all treatment groups, restoring the viability of all three cell types to approximately 80%. These findings indicate that the detoxification capability of cross-linked aptamers (ARMAN) is significantly enhanced when compared to its free aptamer counterparts.

### 3.6. Outstanding Protective Effects of the ARMAN Against RT In Vivo

Detoxification and prevention–detoxification experiments were conducted to evaluate the in vivo detoxification ability of the ARMAN. For the detoxification experiment (Figure 7A), mice were administered tail vein injections of the ARMAN and its counterparts with an equivalent aptamer amount at 15 min, 1 h, and 2 h post RT exposure. Due to the high toxicity of RT, all mice (10/10) in the control group (PBS treatment) died within 4 days of toxin exposure, with a median survival time (MST) of 2 days. RTA and RTB treatment slightly extended the survival of poisoned mice, both with an MST of 5 days. Similar results were observed with the combination therapy of RTA and RTB, which were not statistically different from the single injection of either aptamer. Encouragingly, the administration of the ARMAN led to a distinct prolongation of mouse survival, with 6/10 mice remaining alive 14 days post RT exposure.

For the prevention–detoxification experiment (Figure 7B), the ARMAN and its counterparts were injected 15 min before and 1 and 2 h after Ricin administration. Similarly, all mice (10/10) in the control group died within 4 days. RTA and RTB treatment prolonged the survival of poisoned mice with better detoxification effects compared to the detoxification experiment. As illustrated, 4/10 mice survived in the RTB treated group and 5/10 in the RTB and combination treatment groups. As expected, the administration of the ARMAN also led to the most prominent detoxification effects, with 7/10 mice surviving 14 days post RT exposure.

## 4. Discussion

In this study, we successfully developed a novel aptamer nanoarray (ARMAN), which targets RT by conjugating two high-affinity aptamers specific to the RTA chain and RTB chain onto the surface of GO (Figure 2A). Our findings demonstrate that the ARMAN exhibits enhanced neutralization and protective effects against RT, both in vitro and in vivo, compared to free aptamers. This development holds significant implications for the potential application of ARMAN as a new antidote against Ricin poisoning.

The construction of the ARMAN was based on the rationale that combining aptamers targeting both RTA and RTB could provide synergistic effects in neutralizing Ricin, as previous studies have shown that cocktail therapies targeting multiple subunits of toxins often achieve superior detoxifying effects [11,13,14,15]. By employing click chemistry technology, we successfully cross-linked the two aptamers to GO, creating a stable and biosecure nanoarray. The resulting ARMAN displayed relatively uniform morphology, with an average hydromechanical particle size of approximately 153 nm, as confirmed by the DLS, TEM, and AFM analyses (Figure 3). These characteristics suggest that the ARMAN has good potential for intravenous administration.

It should be mentioned that GO was selected as the nanocarrier for aptamers in this study due to its following advantages for biological applications. GO offers a high surface area, which enables an efficient loading of biomolecules, while its functional groups (e.g., carboxyl, hydroxyl) allow for versatile surface modifications, such as covalent or non-covalent aptamer binding [31,32]. Additionally, GO is highly biocompatible, with studies demonstrating its low toxicity in various biological systems, and it exhibits excellent dispersibility in aqueous environments, promoting stability in physiological conditions [33,34]. These characteristics make GO an ideal choice for applications involving aptamer conjugation.

One of the key advantages of the ARMAN lies in its enhanced affinity for Ricin. As evidenced by our equilibrium dissociation constant (Kd) and off-rate measurements, the ARMAN exhibited significantly higher binding affinity and slower dissociation rate compared to free aptamers (Figure 4). This improved affinity is likely due to the increased local concentration of aptamers on the GO surface, facilitating more efficient interactions with Ricin molecules. Additionally, the large surface area of GO provides ample binding sites for aptamers, further enhancing the overall binding capacity of the ARMAN.

The serum stability of the ARMAN is another critical factor for its potential therapeutic application. Our results showed that the ARMAN maintained excellent stability in an in vitro serum model (Figure 4A). This suggests that the ARMAN is resistant to degradation and aggregation in biological fluids, which is essential for its in vivo efficacy.

The biocompatibility of the ARMAN was also evaluated. In vitro hemolysis assays revealed that the ARMAN had minimal hemolytic activity, even at high concentrations. In vivo blood routine and biochemical examinations showed no significant alterations in hematological and biochemical parameters in mice treated with the ARMAN, indicating its low toxicity and good safety profile. These findings are crucial for the clinical translation of the ARMAN as a therapeutic agent.

Both in and ex vivo experiments demonstrated that the ARMAN exerted excellent protective effects against Ricin-induced cytotoxicity. Compared to free aptamers, the ARMAN significantly increased cell viability in all tested cell lines, restoring it to levels close to the control group. The in vivo protective effects of the ARMAN were further evaluated in a mouse model of Ricin poisoning. Both detoxification and prevention–detoxification experiments showed that the ARMAN significantly prolonged the survival of poisoned mice compared to control groups and mice treated with free aptamers. Notably, in the prevention–detoxification experiment, where the ARMAN was administered both before and after Ricin exposure, 7/10 of mice survived until the end of the observation period. These findings underscore the potential of the ARMAN as an effective antidote against Ricin poisoning, particularly in emergency scenarios where rapid deployment is crucial.

The development of the ARMAN addresses several limitations associated with existing antidotes against Ricin. For instance, mAbs are effective in neutralizing Ricin but suffer from complex preparation processes and long production times, making them impractical for rapid response in emergencies [7,21]. In contrast, aptamers, as chemical antibodies, offer numerous advantages such as ease of synthesis, modification, and scalability [35]. By conjugating aptamers to GO, we further enhanced their efficacy and stability, creating a multifunctional nanoarray with excellent therapeutic potential.

## 5. Conclusions

In conclusion, our study demonstrates that the ARMAN, a novel aptamer nanoarray, exhibits remarkable neutralization and protective effects against RT. With its enhanced affinity, serum stability, biocompatibility, and broad therapeutic efficacy, the ARMAN holds great promise as a new antidote against Ricin poisoning. Future studies should focus on further optimizing the formulation of the ARMAN, as well as exploring its potential applications in other toxin-related emergencies. The successful development of the ARMAN represents a significant step forward in the field of antidote research and has important strategic implications for improving emergency response capabilities.

## Figures and Tables

**Figure 2 pharmaceutics-16-01455-f002:**
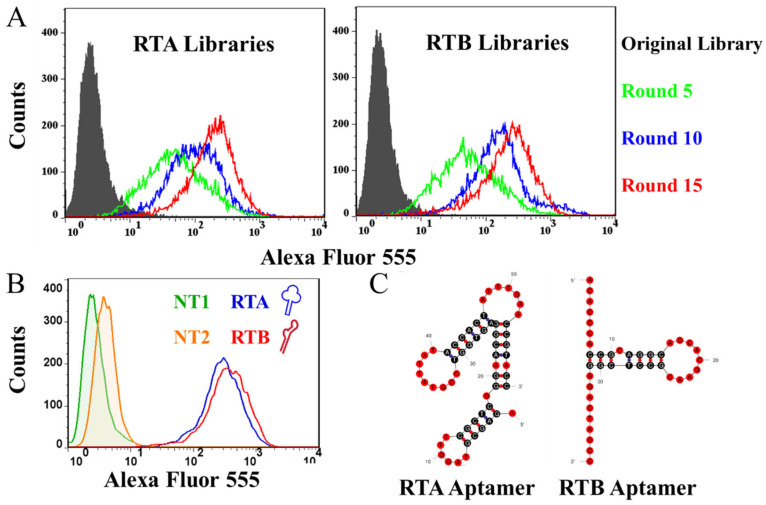
Generation and characterization of anti-RTA and anti-RTB aptamers. (**A**) Binding affinity of aptamer pools with RTA/B coupled microbeads after different round of SELEX by FCM. (**B**) Binding avidity of selected aptamers (RTA and RTB) with RTA/B coupled microbeads by FCM. (**C**) Purported secondary structure of the selected RTA and RTB aptamer.

**Figure 3 pharmaceutics-16-01455-f003:**
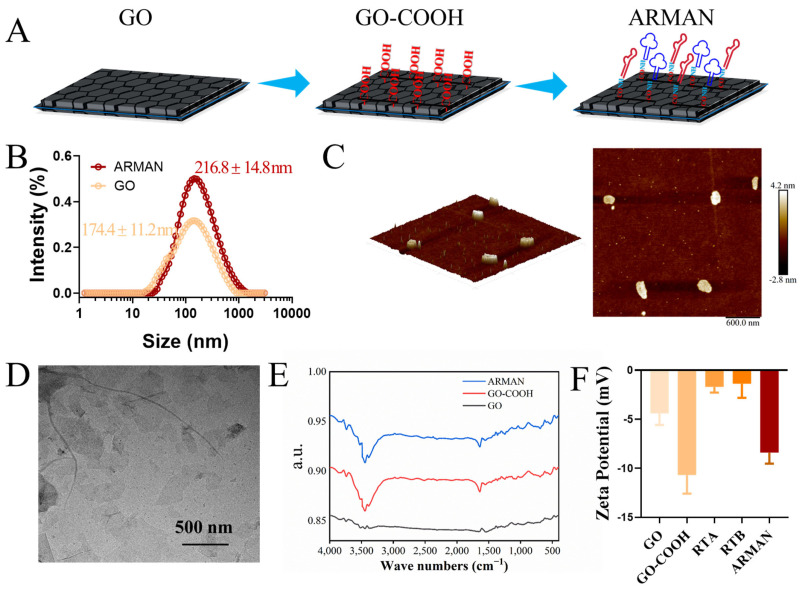
Fabrication and characterization of the ARMAN. (**A**) Schematic diagram of the construction process for the ARMAN, the red and blue symbols represent RTA and RTB specific aptamers, respectively. (**B**) Size distribution of GO and ARMAN determined by DLS. (**C**) AFM morphology of the ARMAN. Scale bar: 600 nm, thickness bar: 0 to 7 nm. (**D**) TEM morphology of the ARMAN. Scale bar: 500 nm. (**E**,**F**) FTIR spectra (**E**) and Zeta potential (**F**) of the ARMAN and counterparts.

**Figure 4 pharmaceutics-16-01455-f004:**
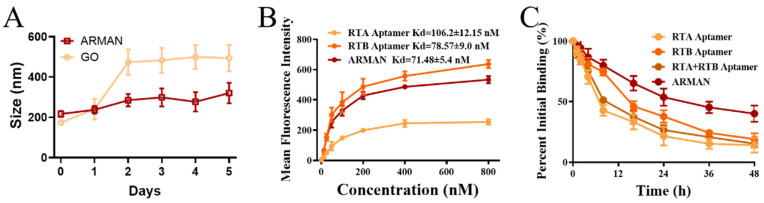
Favorable Serum Stability and Excellent Binding Affinity of the ARMAN. (**A**) Excellent in vitro serum stability of the ARMAN. DMEM containing 50% (*v*/*v*) FBS was employed as an in vitro serum model, and the size distribution of the ARMAN was evaluated by DLS daily. (**B**) RT coupled magnetic beads were respectively incubated with the ARMAN and free aptamers at indicated concentrations from 25 nM to 800 nM, and the MFI was measured by FCM. The dissociation constant (Kd) was calculated based on the concentration MFI curve obtained using IBM SPSS Software (Version 29.0.2). (**C**) Binding off-rate of the ARMAN and parental aptamers from RT coupled magnetic beads. Beads were incubated with fluorescence-labeled aptamer or ARMAN at a concentration of 400 nM. Following incubation, the beads were washed at various time points, and the MFI was assessed by FCM. The proportion of aptamers remaining on the surface of the magnetic beads was subsequently calculated and compared. Data are expressed as means ± SD (*n* = 3).

**Figure 5 pharmaceutics-16-01455-f005:**
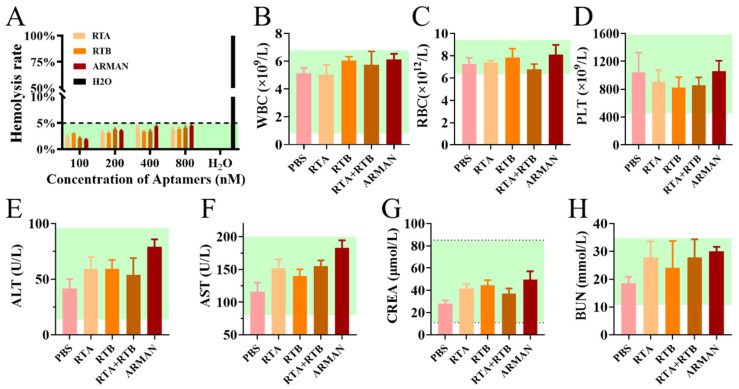
Favorable biocompatibility of the ARMAN and free aptamer counterparts. (**A**) Hemolysis quantification of RBC incubated with the ARMAN and free aptamers. (**B**–**H**) Blood routine and biochemical indexes analysis of mice in plasma samples collected from healthy mice 24 h after tail vein injection of the ARMAN and its counterparts. Data are expressed as means ± SD (*n* = 3).

**Figure 6 pharmaceutics-16-01455-f006:**
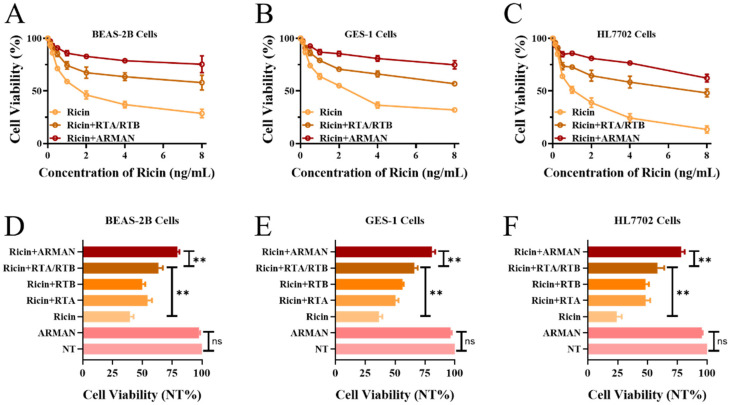
Exceptional protective effects of the ARMAN in vitro. (**A**–**C**) CCK-8 assays were conducted to measure the viability of cells exposed to RT and various antidotes. (**D**–**F**) Cells were incubated with 4 ng/mL of Ricin toxin, and 400 nM of free aptamers and ARMAN were used for detoxification. The cell viability ratios of each treatment group to the control group were calculated and compared. Data are expressed as means ± SD (*n* = 3), ** *p* < 0.01; ns: not significant.

**Figure 7 pharmaceutics-16-01455-f007:**
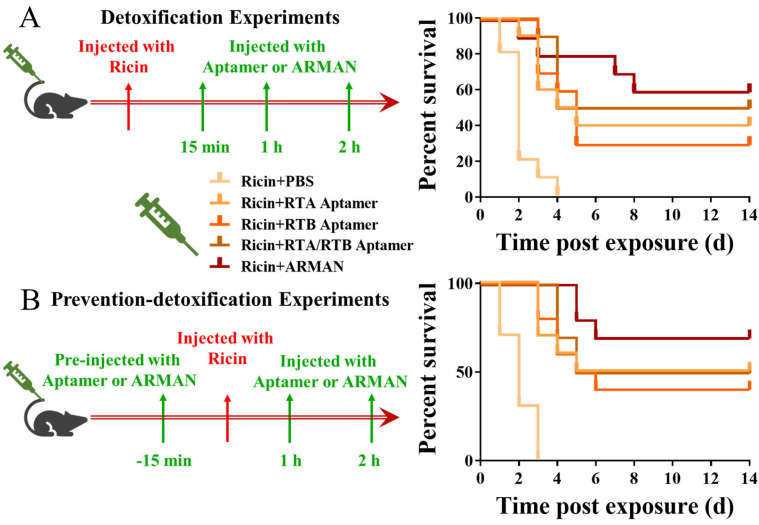
Outstanding protective effects of the ARMAN in vivo. (**A**) For the detoxification experiment, mice were administered tail vein injections of the ARMAN and its counterparts with an equivalent aptamer amount at 15 min, 1 h, and 2 h post RT exposure. (**B**) For the prevention–detoxification experiment, the ARMAN and its counterparts were injected 15 min before and 1 and 2 h after Ricin administration. The mice were closely monitored daily for vital signs until their natural death within 14 days.

**Table 1 pharmaceutics-16-01455-t001:** Sequences of DNA aptamer libraries.

Libraries	Length	Nucleotide Sequence
Library for RTA	60 nt	5′-AGAGC GTAGG TTCGC-*N*30-TATTT GAACC GTACC
Library for RTB	40 nt	5′-ACACC CACCG-*N*20-GAGAC TAGCC

*N*x: A random sequence of x nucleic acids.

## Data Availability

Data are contained within the article.

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
