# Peer review of "Development of a Graphene Oxide-Based Aptamer Nanoarray for Improved Neutralization and Protection Effects Against Ricin"

_pharmaceutics, 2024, doi:10.3390/pharmaceutics16111455_

Round 1
Reviewer 1 Report
Comments and Suggestions for Authors
In this compact, but highly informative work, the authors present a new antidote to such a dangerous chemical warfare agent as ricin. Previously, monoclonal anitibodies (mAB} were proposed for ricin detoxification, which are effective, but have limited use due to their complex and long production time. This makes mABs impractical for rapid response in emergency situations.
The authors used aptamers known as "chemical antibodies" instead of mAB, which, unlike mAB, are easily synthesized and modified. The authors successfully developed a novel aptamer nanoarray (ARMAN) by conjugating two high-affinity aptamers specific to the RTA-chain and RTB-chain of ricin on the surface of graphene oxide (GO). In the experimental part of the work, both the synthesis of GO and the synthesis and optimization of the aptamer composition are described in detail. The structure of ARMAN has been studied by modern physicochemical methods. Its relatively homogeneous morphology with an average size of hydromechanical particles of about 153 nm has been shown, which makes it possible to inject ARMAN intravenously. It has been shown that ARMAN is resistant to degradation and aggregation in biological fluids, for example, in blood serum, which is also important for its therapeutic use.
Other properties of ARMAN, which are important for its medical use, have also been studied in detail: the assessment of its biocompatibility has shown that ARMAN has minimal hemolytic activity even at high concentrations. Routine and in vivo biochemical studies showed no significant changes in hematologic and biochemical parameters in ARMAN-treated mices, suggesting its low toxicity and good safety profile.
The experiments in ex and vivo have shown that ARMAN has excellent protective effects against ricin-induced cytotoxicity. It is noteworthy that ARMAN has a significantly higher protective effect compared to free aptamers, which indicates the existence of a synergy of aptamers associated with GO. Experiments on detoxification and detoxification prevention have shown that ARMAN significantly prolongs the survival of poisoned mices compared to control groups and mices treated with free aptamers. These results highlight the high potential of ARMAN as an effective antidote against ricin poisoning, especially in emergency situations where rapid use of the antidote is crucial.
The authors plan to further optimize the ARMAN formulation, and to study its potential applications in other toxin-related emergencies. The successful development of ARMAN represents a significant step forward in the field of antidote research.
It remains to wish the authors success on this path. I have no comments on the text of the article. However, Fig.1 for some reason, is given in a mirror image. And this needs to be corrected.
.
Author Response
Comments:
The authors plan to further optimize the ARMAN formulation, and to study its potential applications in other toxin-related emergencies. The successful development of ARMAN represents a significant step forward in the field of antidote research.
It remains to wish the authors success on this path. I have no comments on the text of the article. However, Fig.1 for some reason, is given in a mirror image. And this needs to be corrected.
Response: Many thanks for your insightful comments and suggestions, which are very helpful for revising and improving our paper. We feel sorry that something was wrong for Figure 1 in the last submission, which has been corrected in the revised manuscript. Thank you once again for your valuable feedback, which will help us strengthen our manuscript.
Reviewer 2 Report
Comments and Suggestions for Authors
The authors presented a novel study on ricin detoxification using an aptamer-based nanoarray, describing a complete technological path from aptamer synthesis and nanoarray development to in vivo studies. The results and discussion section clearly show the efficiency and biocompatibility of the proposed nanoarray. However, I have a few uncertainties that I will present here:
1. Authors enlisted only biologically relevant materials and cell lines in sub-section 2.1 (lines 98-106). Could they provide a full list of materials used in this study?
2. Some information is lacking in section 2.5 "Preparation and characterization of ARMAN", such as the obtained GO-COOH concentration and sonication time. Was the GO-COOH dispersion in PBS prepared from powder GO-COOH or water dispersion GO-COOH, because the sentence in section 2.5, line 152 is not clear. Additionally, the final aptamer concentration in the ARMAN dispersion is not given in this section, while it appears in the following sections: 2.7 and 2.10; please, add this information to align with the following text.
3. The authors synthesized aptamers for the RT-A and RT-B chains using the standard SELEX method, which is clearly explained in the manuscript. However, in the literature, there are already established aptamers sequences for both chains of ricin:
- Ricin A-chain RNA aptamer: L. Qi et al., Sensors and Actuators B: Chemical, 314 (2020), 128073. https://doi.org/10.1016/j.snb.2020.128073
- Ricin B-chain DNA aptamer: E.A. Lamont et al., Analyst, 136 (2011), 3884-3895. https://doi.org/10.1039/C1AN15352H .
Why did the authors decide to develop new sequences for both ricin chains? Perhaps giving more information on the existing aptamers in the introduction (1. Introduction, lines 76-88) and demand for new aptamer sequences synthesis in the results or discussion.
Please note that Figure 1 is mirror-inverted, making it difficult to interpret.
4. May I ask the authors to provide the information on the GO flake (non-carboxylated) (commercial/in-house prepared and average lateral size) before ARMAN construction?
5. Characterization of the ARMAN part (Section 3.2, Successful fabrication of ARMAN) could provide more information. For example:
- TEM and AFM micrographs do not show the size values (length measurement marks) as claimed in the text (~ 200 nm). The AFM micrograph shows the thickness of the ARMAN, yet it is not explained in the text; it could give interesting information about ARMAN morphology. Please, make the scale bar visible on the AFM micrograph and normalize the height bar to 0.
- The FTIR spectra indicate differences between GO and GO-COOH/ARMAN. However, there is no distinct difference between GO-COOH and ARMAN, which may suggest aptamer binding (amide bond formation) that supports other characterization results. Additionally, the y-axis of the FTIR comparison graph should be labeled in arbitrary units rather than absolute units. The FTIR measurements could also be explained in more detail (sample preparation) in section 2.5 (Preparation and characterization of ARMAN), lines 169-171.
- Zeta potential measurements indirectly confirm aptamer binding to the functionalized GO through the observed change in values; however, the reason for this change is not clearly explained in the text. A plausible explanation could be the neutralization of COOH groups due to aptamer binding via click chemistry (amide bond formation). However, as aptamers also carry a negative charge from their phosphate backbone, indicated by the negative Zeta potential of the free-standing aptamer, a more detailed explanation of the Zeta potential change is necessary to support the statement in lines 282-285.
6. The efficiency of ARMAN compared to the free-standing aptamers in both in vitro and in vivo experiments is very well presented, with authors explaining that aptamers anchored on the surface of GO are locally more concentrated than free aptamers and enable better detoxification properties. A minor suggestion is to have the same y-axis step in Fig. 7 graphs for easier data comparison. Could the authors provide more information on why they chose GO as a nanocarrier for aptamers instead of other nanomaterials with similar properties, such as high surface area and biocompatibility? What gives the advantage to GO over other nanomaterials studied for similar applications? I suggest introducing GO as the material of choice in the Introduction section to provide supporting context.
7. Please revise references. References 7 and 8 are the same.
Author Response
General comments:
The authors presented a novel study on ricin detoxification using an aptamer-based nanoarray, describing a complete technological path from aptamer synthesis and nanoarray development to in vivo studies. The results and discussion section clearly show the efficiency and biocompatibility of the proposed nanoarray.
Comments 1: Authors enlisted only biologically relevant materials and cell lines in sub-section 2.1 (lines 98-106). Could they provide a full list of materials used in this study?
Response: Thank you very much for your suggestion. All the relevant materials employed in this study were listed in the descriptions of the relevant experimental methods, along with the detailed supplier information. We have thoroughly reviewed the entire manuscript and added any missing information as needed. Thank you again for your kind suggestion.
Comments 2: Some information is lacking in section 2.5 "Preparation and characterization of ARMAN", such as the obtained GO-COOH concentration and sonication time. Was the GO-COOH dispersion in PBS prepared from powder GO-COOH or water dispersion GO-COOH, because the sentence in section 2.5, line 152 is not clear. Additionally, the final aptamer concentration in the ARMAN dispersion is not given in this section, while it appears in the following sections: 2.7 and 2.10; please, add this information to align with the following text.
Response: Thank you very much for your suggestion. We included specific information regarding the concentration of GO-COOH and the sonication time. The GO-COOH was redispersed in PBS at an approximate concentration of 2 mg/mL and sonicated for about 30 minutes until the supernatant appeared light brown with no visible solid particles. We also added the final aptamer concentration in the ARMAN solution, which ranges from approximately 20 to 29 mM across different batches, ensuring consistency with the information provided in subsequent sections. All thses modifications have been included in the revised manuscript in accordance with your suggestions. Thank you for your suggestion.
Comments: The authors synthesized aptamers for the RT-A and RT-B chains using the standard SELEX method, which is clearly explained in the manuscript. However, in the literature, there are already established aptamers sequences for both chains of ricin:
- Ricin A-chain RNA aptamer: L. Qi et al., Sensors and Actuators B: Chemical, 314 (2020), 128073. https://doi.org/10.1016/j.snb.2020.128073
- Ricin B-chain DNA aptamer: E.A. Lamont et al., Analyst, 136 (2011), 3884-3895. https://doi.org/10.1039/C1AN15352H .
Why did the authors decide to develop new sequences for both ricin chains? Perhaps giving more information on the existing aptamers in the introduction (1. Introduction, lines 76-88) and demand for new aptamer sequences synthesis in the results or discussion.
Please note that Figure 1 is mirror-inverted, making it difficult to interpret.
Response: Thank you very much for your suggestion. We appreciate your insightful comments regarding the development of new aptamer sequences for the ricin A and B chains. In our research, we opted not to use the existing sequences mentioned in the literature, as we aimed to develop aptamer sequences that would provide us with independent intellectual property rights. Besides, most existing aptamer research has primarily centered on the diagnosis of ricin exposure rather than the development of therapeutic agents aimed at neutralizing the toxin. This gap highlights the need for novel aptamer sequences that can provide effective solutions for ricin detoxification. Additionally, our research platform has a mature technology for the selection of aptamers, allowing us to efficiently screen for new sequences without requiring significant manpower or time. We believe that developing our unique aptamer sequences will enhance the versatility and applicability of our work. Besides, we have included more detailed information about existing aptamers in the revised manuscript according to your suggestion. We feel sorry that something was wrong for Figure 1 in the last submission, which has been corrected in the revised manuscript. Thank you once again for your valuable feedback, which will help us strengthen our manuscript.
Comments 4: May I ask the authors to provide the information on the GO flake (non-carboxylated) (commercial/in-house prepared and average lateral size) before ARMAN construction?
Response: Thank you very much for your suggestion. The non-carboxylated GO was commercially bought from Ashine new carbon material Changzhou Co., LTD (Jiangsu, China), and the average lateral size is almost the same as carboxylated GO. We did not perform any additional treatment on the particle size of GO. The source of GO was added in the revised manuscript. Thank you again for your kind suggestion.
Comments 5: Characterization of the ARMAN part (Section 3.2, Successful fabrication of ARMAN) could provide more information. For example:
- TEM and AFM micrographs do not show the size values (length measurement marks) as claimed in the text (~ 200 nm). The AFM micrograph shows the thickness of the ARMAN, yet it is not explained in the text; it could give interesting information about ARMAN morphology. Please, make the scale bar visible on the AFM micrograph and normalize the height bar to 0.
- The FTIR spectra indicate differences between GO and GO-COOH/ARMAN. However, there is no distinct difference between GO-COOH and ARMAN, which may suggest aptamer binding (amide bond formation) that supports other characterization results. Additionally, the y-axis of the FTIR comparison graph should be labeled in arbitrary units rather than absolute units. The FTIR measurements could also be explained in more detail (sample preparation) in section 2.5 (Preparation and characterization of ARMAN), lines 169-171.
- Zeta potential measurements indirectly confirm aptamer binding to the functionalized GO through the observed change in values; however, the reason for this change is not clearly explained in the text. A plausible explanation could be the neutralization of COOH groups due to aptamer binding via click chemistry (amide bond formation). However, as aptamers also carry a negative charge from their phosphate backbone, indicated by the negative Zeta potential of the free-standing aptamer, a more detailed explanation of the Zeta potential change is necessary to support the statement in lines 282-285.
Response: Thank you for your insightful comments, which have greatly improved our manuscript. Below, we address each point in turn:
- TEM and AFM Micrographs: We have revised the TEM and AFM images to include clear scale bars, accurately reflecting the particle size as described in the text. Additionally, the AFM micrograph has been modified to normalize the height bar to 0, providing a clearer representation of ARMAN’s thickness. This information has been incorporated into the revised text to offer a more comprehensive description of ARMAN’s morphology.
- FTIR Spectra: We appreciate the reviewer’s suggestion regarding the FTIR y-axis. The comparison graph has been updated to display the y-axis in arbitrary units, ensuring a more standardized presentation. While GO-COOH and ARMAN show subtle spectral differences, we acknowledge that FTIR alone may not reveal distinct aptamer binding signals. This can be attributed to the fact that the FTIR characterization of amide bond formation can be challenging due to peak overlaps with existing functional groups. Specifically, both carboxyl and amide groups exhibit absorption bands in similar regions, complicating spectral differentiation. Despite these limitations in the FTIR spectra, the binding between the aptamers and GO-COOH may still be inferred indirectly by size or morphological alterations. All there have been added in the revised manuscript. We also provided additional details on sample preparation in Section 2.5 to improve reproducibility.
- Zeta Potential Explanation: We have expanded the text to describe the dual effect contributing to the Zeta potential shift. This shift results from the partial neutralization of COOH groups due to amide bond formation, along with the residual negative charge from the aptamer’s phosphate backbone. This detailed explanation strengthens the evidence for successful aptamer binding.
All these revisions have been added in the revised manuscript. Thank you again for your suggestions.
Comments 6: The efficiency of ARMAN compared to the free-standing aptamers in both in vitro and in vivo experiments is very well presented, with authors explaining that aptamers anchored on the surface of GO are locally more concentrated than free aptamers and enable better detoxification properties. A minor suggestion is to have the same y-axis step in Fig. 7 graphs for easier data comparison. Could the authors provide more information on why they chose GO as a nanocarrier for aptamers instead of other nanomaterials with similar properties, such as high surface area and biocompatibility? What gives the advantage to GO over other nanomaterials studied for similar applications? I suggest introducing GO as the material of choice in the Introduction section to provide supporting context.
Response: Thank you for your helpful suggestions. As for the y-axis step uniformity in Figure 7, we attempted to combine both graphs to use a unified y-axis scale, but found that the increased data and line density created a visually cluttered effect, detracting from the clarity of the individual datasets. Therefore, we have chosen to retain the original separate display format, which provides clearer data visualization. We appreciate your understanding and are open to any further adjustments if necessary.
Besides, GO was selected as the nanocarrier for aptamers in this study due to its unique combination of properties that make it advantageous for biological applications. GO offers a high surface area, which enables efficient loading of biomolecules, while its functional groups (e.g., carboxyl, hydroxyl) allow for versatile surface modifications, such as covalent or non-covalent aptamer binding. Additionally, GO is highly biocompatible, with studies demonstrating its low toxicity in various biological systems, and it exhibits excellent dispersibility in aqueous environments, promoting stability in physiological conditions. These characteristics make it an ideal choice for applications involving aptamer conjugation. In addition, we acknowledge that various other two-dimensional (2D) nanomaterials—such as boron nitride (BN), DNA etc.—also possess advantageous properties, which could make them suitable for similar applications. These materials show promise in biomolecule immobilization and delivery, and they each offer unique electronic, optical, and chemical characteristics that may further enhance the detoxification and targeting capabilities of aptamer-based systems. In future experiments, we plan to explore these alternative 2D nanomaterials as carriers for aptamers to evaluate their effectiveness and determine the optimal material for advancing this research. This comparative study will help us identify the most efficient nanocarrier for potential therapeutic applications. The above-mentioned information has been added in the “discussion section” in the revised manuscript. Thank you again for your kind suggestions.
Comments 7: Please revise references. References 7 and 8 are the same.
Response: Reference 8 has been deleted in the revised manuscript. Thank you very much for your suggestion.

Reviewer 3 Report
Comments and Suggestions for Authors
1. Figure 1 should be flipped upside down to be readable in the provided PDF file.
2. Detailed information about the urea-formaldehyde magnetic microbeads used (including the bead size, concentration, etc.) and the protocols for immobilization of ricin to the surface of such magnetic microbeads should be added.
Author Response
Comments 1: Figure 1 should be flipped upside down to be readable in the provided PDF file.
Response: We feel sorry that something was wrong for Figure 1 in the last submission, which has been corrected in the revised manuscript. Thank you very much for your consideration.
Comments 2: Detailed information about the urea-formaldehyde magnetic microbeads used (including the bead size, concentration, etc.) and the protocols for immobilization of ricin to the surface of such magnetic microbeads should be added.
Response: Thank you for your valuable feedback. We have provided a comprehensive description of the protocols used for immobilizing RTA/B onto the surface of these magnetic microbeads. These details have been incorporated into the revised manuscript to enhance clarity and reproducibility. We appreciate the reviewer’s input, which has helped to improve the rigor of our experimental methodology.
